# Peptide-Bound Glycative, AGE and Oxidative Modifications as Biomarkers for the Diagnosis of Alzheimer’s Disease—A Feasibility Study

**DOI:** 10.3390/biomedicines12092127

**Published:** 2024-09-19

**Authors:** Anne Grosskopf, Jette Rahn, Ahyoung Kim, Gábor Szabó, Dan Rujescu, Frank Klawonn, Andrej Frolov, Andreas Simm

**Affiliations:** 1Clinic for Cardiac Surgery, University Medicine Halle, Martin Luther University Halle-Wittenberg, 06120 Halle (Saale), Germany; 2Department of Bioorganic Chemistry, Leibniz Institute of Plant Biochemistry, 06120 Halle (Saale), Germany; 3Department of Psychiatry, Psychotherapy, Psychosomatic Medicine, Martin Luther University Halle-Wittenberg, 06112 Halle (Saale), Germany; 4Department of Psychiatry and Psychotherapy, Medical University of Vienna, 1090 Vienna, Austria; 5Biostatistics Group, Helmholtz Centre for Infection Research, 38124 Braunschweig, Germany; 6Laboratory of Analytical Biochemistry and Biotechnology, Timiryazev Institute of Plant Physiology, 127276 Moscow, Russia

**Keywords:** Alzheimer’s disease, advanced glycation end products (AGE), nanoLC-LIT-Orbitrap-MS, cerebrospinal fluid, biomarker

## Abstract

**Background:** The diagnosis of Alzheimer’s disease (AD) relies on core cerebrospinal fluid (CSF) biomarkers, amyloid beta (Aβ) and tau. As the brain is then already damaged, researchers still strive to discover earlier biomarkers of disease onset and the progression of AD. Glycation, advanced glycation end products (AGEs) and oxidative modifications on proteins in CSF mirror the underlying biological mechanisms that contribute to early AD pathology. However, analyzing free AGEs in the body fluids of AD patients has led to controversial results. Thus, this pilot study aimed to test the feasibility of detecting, identifying and quantifying differentially glycated, AGE or oxidatively modified peptides in CSF proteins of AD patients (*n* = 5) compared to a control group (*n* = 5). **Methods:** To this end, we utilized a data-dependent (DDA) nano liquid chromatography (LC) linear ion trap-Orbitrap tandem mass spectrometry (MS/MS) ) approach and database search that included over 30 glycative and oxidative modifications in four search nodes to analyze endogenous modifications on individual peptides. Furthermore, we quantified candidate peptide abundance using LC Quan. **Results:** We identified 299 sites of early and advanced glycation and 53 sites of oxidatively modified tryptophan. From those, we identified 17 promising candidates as putative biomarkers (receiver operating curve-area under the curve (ROC-AUC) > 0.8), albeit without statistical significance. **Conclusions:** The potential candidates with higher discrimination power showed correlations with established diagnostic markers, thus hinting toward the potential of those peptides as biomarkers.

## 1. Introduction

The brain disease pathology of Alzheimer’s disease (AD) is characterized by amyloid plaques, axonal degeneration and intra-neuronal tangles, which can be monitored using the core cerebrospinal fluid (CSF) biomarkers amyloid beta peptides 1–40 (Aβ_40_), amyloid-beta 1–42 (Aβ_42_), total tau (t-tau) and phosphorylated tau (p-tau) proteins as well as Aβ positron emission tomography (PET) imaging [1]. A CSF profile of biochemical markers suggestive of AD is associated with low Aβ_42_ levels combined with high t-tau and p-tau levels and, in this combination, sufficient to diagnose AD [2].

The first asymptomatic stage in the AD continuum, preclinical AD, lasts, on average, 6 to 10 years. Already 5 to 10 years before that, tangle formation and Aβ deposition are detectable via CSF biomarkers [3]. However, those markers already resemble pathological changes in the brains of the patients. Although CSF core biomarkers (Aβ_42_, t-tau, p-tau) are already indicative in preclinical AD [4], there are still limitations, such as high correlations of p-tau, t-tau and tau PET, which limit the amount of independent diagnostic information available [5], and the need of CSF for detection.

Apart from the core amyloid (A) and tau (T) fluid biomarkers for diagnostics, additional markers are currently investigated and considered, especially for staging, prognosis and identification of co-pathologies [1].

Markers of neuronal injury, dysfunction and degeneration, summarized as N-group, include neurofilament light chain (NEFL), Visinin-like protein-1 (VILIP-1), synaptosomal-associated protein-25 (SNAP-25) and neurogranin [6,7]. NEFL levels, for example, reflect white matter axonal damage in various neurological diseases and dementias and are highly clinically relevant for staging and progression [8].

Another group of emerging protein markers is related to brain inflammation (I) mediated by astrocytes and microglia. It includes interferons, soluble TREM2 (sTREM2) and glial fibrillary acidic protein (GFAP) as well as chitinase-like 3 protein 1 (YKL40) [1,6]. GFAP and YKL40 are involved in astrogliosis, mediating early disease progression [9]. Apart from the mentioned groups, additional markers for vascular damage (V) or synucleinopathy (S) are established to provide information on relevant co-pathologies [1].

Preceding inflammation and neuronal injury, increased oxidative stress by impaired mitochondrial function is described in aging and neurodegenerative diseases like AD [10]. Furthermore, it mechanistically links many described risk factors of AD-like cardiovascular diseases (CVD), insulin resistance, obesity and type 2 diabetes mellitus (T2DM) [11,12,13,14]. Furthermore, it was established that a deregulated glucose metabolism is inherent for both early AD and T2DM. Indeed, recent proteomics approaches to uncover early biomarkers of AD in CSF revealed prominent changes in enzymes of glucose metabolism, but there are no established biomarkers yet [15,16,17,18].

It is feasible to assume that damage to proteins induced by reactive oxygen species (ROS) or sugar molecules via the Maillard reaction could be detected even before changes in protein abundances, similar to hemoglobin A1c (HbA1c) in the onset and monitoring of diabetes. Both glucose metabolism deregulation and ROS can facilitate stable modifications on CSF proteins, which could be indicative of the very early disease mechanisms in AD.

Advanced glycation end products (AGEs) are the stable end-stage adducts of the reaction of carbohydrates, e.g., glucose, fructose or ribose, or reactive dicarbonyls, like glyoxal (GO) or methylglyoxal (MGO), with free amino or guanidine groups of proteins, mainly side chains of lysine, arginine and, to a lesser extent, histidine or cysteine [19]. The accumulation of AGEs and subsequent protein crosslinking can accelerate pathological processes, like amyloidosis and tangle formation [20,21], and is independently associated with worse cognition and mild cognitive impairment (MCI) [22,23].

Both tau and Aβ are known to be glycated and subsequently AGE modified [24,25,26,27], and glycation exacerbates the neurotoxicity of β-amyloid [26,28]. Apart from those proteins, several studies demonstrated changes in general CSF levels of Amadori products, N(6)-carboxymethyl lysine (CML) and other AGEs, e.g., pentosidine, but also of nitration and oxidation products in patients with AD and vascular dementia [29,30,31,32]. Recently, Carvalho and Moreira demonstrated that CML induces neuronal damage in vivo [33]. However, AGE modifications were also found in CSF independent of pathologies [34].

In an interplay between glucose metabolism aberrations, AGE formation and oxidative stress, oxidative and glycoxidative modifications on peptides are also described. While oxidation on methionine is an unstable modification prone to artifacts in mass spectrometric analysis, oxidation of tryptophan to different oxidation products like kynurenine was previously described as stable and capable of monitoring ROS in respiring tissues [35,36]. Oxidative stress can then amplify and modify AGE formation.

Thus, identifying glycated, AGE and oxidatively modified proteins in CSF samples of patients with AD would allow us to monitor the earliest changes in oxidative stress and glucose metabolism on a molecular level and could be the basis for identifying biomarkers or therapeutic targets.

In this feasibility study, we aim to identify and quantify endogenous, stable glycated, AGE and oxidatively modified peptides from CSF proteins of AD patients compared to a control group without prior enrichment. Additionally, we intend to determine their potential as early biomarkers for diagnosing AD.

## 2. Materials and Methods

### 2.1. Patients, CSF Sample Collection, and Sample Handling

CSF samples were collected from patients between 2014 and 2016 following a standardized protocol during a stay at the University Hospital and Polyclinic for Psychiatry, Psychotherapy and Psychosomatics of the University Medicine Halle (Saale). CSF was obtained by lumbar puncture as follows. The patient’s back was sterilized, covered with a sterile cloth with a hole and sterilized again. At first, an introducer was applied to the puncture site. Subsequently, the spinal canal was punctured horizontally with a slight cranial direction using an atraumatic 22G LP-needle type Sprotte. After removal of the stylet, CSF was collected by a qualified study nurse in six numbered 10 mL polypropylene screw cap tubes. The first three tubes were used for diagnostics. The CSF samples (2 mL each) in the other three tubes were stored in an ice-cooled box and stored at −80 °C immediately after the end of the procedure until further use for research purposes without centrifugation. Patient data were pseudonymized.

Samples for this study (*n* = 10) were selected from the available samples and grouped based on the diagnosis of AD and the control group presenting with other neurological diseases (OND) according to the patient’s discharge diagnosis made by a qualified physician. Other neurological diseases permitted as controls were pain disorders, schizophrenia, depressive disorders and delirium. Patients with other types or mixed forms of dementia and with evident cognitive impairment (mini mental state examination (MMSE) < 24) were excluded from the control group to minimize the overlap with AD pathology and biomarkers in this feasibility study. For the AD group, physician-diagnosed AD was a prerequisite, as was cognitive impairment (MMSE < 24). Furthermore, patients with mixed pathologies were excluded. Patient characteristics are presented in Table 1.

After thawing on ice, the CSF samples were aliquoted into 1.5 mL polypropylene tubes and directly subjected to protein determination via Pierce 660 nm Protein Assay (#22660, Thermo Fisher Scientific, Waltham, MA, USA) and then enzymatic digestion.

### 2.2. Determination of Diagnostic Parameters Amyloid Beta and Tau

The amyloid beta peptides with 40 (Aβ_40_) and 42 amino acids (Aβ_42_) as well as phosphorylated tau (p-tau) and total tau (t-tau) were determined by the amedes MVZ Laboratoriumsdiagnostik und Mikrobiologie Halle/Leipzig (Halle, Germany) using the solid phase enzyme immunoassays INNOTEST β-Amyloid(1-40), INNOTEST β-Amyloid(1-42), INNOTEST PHOSPHO TAU(181P), and INNOTEST hTAU Ag ELISAs (all FujireBio, Gent, Belgium). The results and applicable diagnosis cut-offs are presented in Table 1. Aβ-ratio (Aβ_42_ × 10/Aβ_40_) and a discrimination line β-amyloid (Aβ = 150 + 0.73 × t-tau) were calculated subsequently for evaluation in this study.

### 2.3. Mini-Mental State Examination (MMSE)

MMSE (German translation) was routinely performed during patient assessment [37]. The results were reported as a summed score.

### 2.4. Enzymatic Digestion

Enzymatic protein digestion was conducted as described by Mamontova and colleagues [38] with some changes. In detail, 125 µg protein of CSF samples was diluted 10-fold in shotgun buffer (100 mmol/L Tris-HCL, pH 7.2, 8 mol/L urea, 2 mol/L thiourea). Subsequently, 35 µg was further diluted in shotgun buffer containing 0.1% (*w*/*v*) anionic acid labile surfactant II (AALSII, Protea Biosciences, Morgantown, WV, USA) to a final concentration of 0.0825% AALSII. Subsequently, TCEP (tris(2-carboxyethyl)phosphine) in shotgun buffer was added to a final concentration of 5.5 mmol/L and vortexed, and the samples were incubated for 30 min at 37 °C while shaking. After cooling to room temperature, chloroacetamide was added to a final concentration of 11 mmol/L, followed by an incubation for one hour at 4 °C in the dark. Before digestion, the samples were diluted 9-fold in ammonium bicarbonate buffer (100 mmol/L, pH 8.0). Then, trypsin (Trypsin NB sequencing grade, Serva, Heidelberg, Germany) was added at a protease to protein ratio of 1:20 and, after five hours of incubation at 37 °C, was added again at a protease to protein ratio of 1:40 for an additional 12 h of digestion. Finally, 5 µg of protein was aliquoted for digestion control on SDS-PAGE. The digestion was repeated if the digestion control showed visible bands after Coomassie staining.

To degrade the AALSII, trifluoroacetic acid (TFA) was added to obtain a final concentration of 1% (*v*/*v*), and the samples were incubated for 30 min at 37 °C while shaking.

### 2.5. Solid Phase Extraction

As previously described, solid phase extraction and desalting of peptides were performed [39]. In short, stage tips were prepared with six layers of C18 material, conditioned and loaded with a maximum of 30 µg digest. Then, peptides were washed (0.1% formic acid (FA)) and eluted. Instead of a 1-step elution of the original protocol, three elution steps with 40%, 60% and 80% (*v*/*v*) acetonitrile (AcN) in 0.1% FA were utilized, as described by Spiller and colleagues [40]. Subsequently, peptides were dried under vacuum application.

### 2.6. Nano Liquid Chromatography (nLC)-Linear Ion Trap (LIT)-Orbitrap-Tandem Mass Spectrometry (MS/MS)

Digests of each sample containing 500 ng of peptides were dissolved in 10 µL of 3% (*v*/*v*) AcN and 0.1% TFA. Subsequently, the samples were loaded onto aC18 trap column (0.3 mm × 5 mm, 3 µm particles, Acclaim PepMap 100) at a flow rate of 30 µL/min for 15 min and separated at a flow rate of 300 nL/min on another C18 column (75 µm × 250 mm, 2 µm particles, EASY-Spray PepMap 100). The workflow used was an nLC 1000 nano-HPLC system coupled to a hybrid LIT Orbitrap Velos Pro mass spectrometer via an EASY-Spray™ Source (all Thermo Fisher Scientific, Bremen, Germany). The linear gradient was built from eluents A (0.1% FA in water) and B (0.08% FA in AcN), ramping from 1 to 35% B for 90 min and 35 to 85% eluent B for an additional 5 min. After the gradient, the column was washed (5 min) and re-equilibrated to 1% B (10 min). Data-dependent acquisition (DDA) experiments were carried out in a positive ion mode. The analysis method contained an Orbitrap-MS survey scan and Top5 MS/MS for 5 s. Considered charge stages were +2 to +6; unassigned and +1 charge states were rejected. The analysis was run with the following settings: first spectrometer (MS1) resolution at 60,000; ion spray voltage at 1.9 kV; capillary temperature 275 °C; MS1 AGC target at 3 × 10^−6^; MS1 max injection time—1 ms; *m*/*z* scan range 300–1500; Profile spectrum data type for MS1; fragmentation was performed by collision activated dissociation (CID) at normalized energy 35 V; second spectrometer (MS2) resolution 17,500; AGC target for MS2 was 5 × 10^−4^; MS2 max injection time 50 ms; MS2 isolation window 2 *m*/*z*; automatic scan range; Centroid spectrum data type for MS2; intensity threshold for fragmentation—3 × 10^−4^; exclude isotopes was enabled; and duration of dynamic exclusion was 60 s.

### 2.7. Database Search

Identification of peptides and annotation of proteins relied on a search against the Homo sapiens protein database. (Taxon ID: 9606, 152,269 entries, downloaded from Uniprot, 24 July 2018). The Sequest search engine and Proteome Discoverer 2.2 software (PD) (Thermo Fisher Scientific, Bremen, Germany) were used. Variable modifications and database search parameters are summarized in Appendix A.

### 2.8. Peak Quantification

All identified peptides from the database search containing at least one glycative, AGE or oxidative modification with high or medium confidence were exported from PD with the respective *m*/*z* values and retention times. Peptides containing only various oxidations were not considered candidates because of their instability. The exported peptides were then numbered for a more straightforward evaluation. Individual peptides were quantified by characteristic extracted ion chromatograms (*m*/*z* ± 0.03 Da) at the retention times specified by the database search with correction of the retention times (t_Rs_) to the peak apex values within a 60 s window. The corresponding peaks of extracted peptide signals were manually integrated using LC Quan 3.0 (Thermo Fisher Scientific, Bremen, Germany) in batch mode. Short, standard settings for the quantification algorithm ICIS were used except for the base window, which was set to 10. An explore method of identification with *m*/*z* values and retention times was created for each candidate to search for a matching peak in all raw data files via the quantitate window workflow. If a peak could be assigned, the areas under the curves (AUC) of the peak chromatograms in the individual data files were reported as a response and exported for further evaluation. During peak detection, peaks of the desired peptide *m*/*z* value represented only by a shoulder of an unrelated, higher peak were excluded from further analysis. Also, spectra with insufficient annotation in PD—less than five individual amino acid (aa) signals assigned—led to the peptide being excluded from the candidate list. Spectra of the candidates are assembled in Appendix A.

### 2.9. Statistical Evaluation

Peptides with the derived AUCs were considered for further analysis if the corresponding signals were present in at least three of five replicate samples of both groups. Furthermore, peptides with two signals in one group but more than three in the second were also considered. Missing data were not imputed unless described otherwise. The area under the receiver operating curve (ROC-AUC) was calculated for each peptide as described previously based on the selection of the 20 best attributes for classifier training and a leave-one-out (jackknife) cross-validation strategy [41,42]. The results were utilized to estimate the potential of the candidate peptides as prospective disease-related biomarkers for AD versus OND. Due to the small cohort size and the large number of peptides considered, after any correction for multiple testing, even a perfect ROC-AUC value of 1 would no longer have statistical significance. Thus, no correction for multiple testing was carried out in this analysis. However, even without statistical significance, AUC values can help determine the candidate peptides with the best potential to serve as biomarker candidates. One way to evaluate whether a small dataset contains potential biomarkers is by calculating the high abundance AUC analysis (HAUCA) [42]. To generate the high HAUCA curves to evaluate the enrichment of discriminating peptides, missing values were substituted by the median value of the respective groups (OND or AD), and Bonferroni–Holm correction for multiple testing was applied [42,43].

Additionally, Spearman’s rank correlations were calculated for all peptide candidates and the respective patients’ diagnostic biochemical parameters (see Table 1) and age.

All statistical analyses were carried out with R software version 4.2.2. The R packages used were “pROC” by Robin and colleagues for ROCS-AUCs [44] and “Hmisc” for the imputation of missing values. The HAUCA analysis was recently included in the R shiny app “HiPerMAb” [41].

## 3. Results

### 3.1. Patient Characteristics

CSF samples from 10 patients either presenting with AD and cognitive decline or with OND without cognitive impairment or dementia as a “control group” were selected. The patient characteristics (Table 1) varied significantly in their MMSE score, which was the primary selection criterion apart from Alzheimer’s diagnosis in this pilot study. Furthermore, the sex and age distribution varied significantly between the groups. The median age in the OND group was 63 years, while it was 80 years in the AD group. The mean values of amyloid-beta (Aβ) and tau markers were not significantly different but adhered mostly to the cut-off values reported for diagnosis (Table 1).

### 3.2. Endogenous, Peptide-Bound Glycative, AGE and Oxidative Modifications Are Detectable in CSF and Convey Peptide-Specific Information

The analysis revealed that multiple sites of early glycation, intermediate products, advanced glycation end products (AGEs) and oxidative modifications could be detected in the CSF of patients or controls at endogenous levels without prior depletion of high-abundant proteins (Figure 1).

In total, 299 sites of glycative, AGE and oxidative modifications on 219 peptides could be identified based on the database search output of the MS2 analysis (Appendix A). Lysine (K) and arginine (R) modifications were found with a similar frequency of 124 and 122 sites, respectively, and additionally, 53 sites of tryptophan side-chain oxidation were detected. Those modifications initiated by ROS included 31 kynurenine, 13 hydroxy-kynurenine and nine oxolactone modification sites.

Early glycative modifications were generally detected at lower rates, while higher numbers of sites were observed for intermediate and advanced structures. However, the data provide evidence of transient reaction of trioses, tetroses, pentoses, presumably ribose and fructose, as well as glyoxal (GO) with CSF proteins as early glycative modifications. This indicates the presence and change in abundance of those sugars and carbonyls, probably through changing metabolic pathways like glycolysis or the pentose phosphate pathway. The most common chemically labile intermediates detected were GO or glyceraldehyde (GA)-derived hydroimidazolinones, Glarg (18) and Glap (12), which can react further, forming carboxymethyl (CM)-AGEs. 3-deoxyglucosone intermediates (3-DG) are fructose-derived precursors of pyralline, while Lederers glucosone and pentosone, which were also detected, can react to glucosepane and pentosidine, respectively. The most common classical, single AGE modification was a carboxyethyl(CE)-moiety on K or R, followed by CML/A (Figure 1). Apart from pyrazine, detected only six times and derived from GA, all other stable AGEs are end products of GO and MGO reactions. Apart from those, many different glycoxidative amide AGEs formed from sugars only in oxidative conditions were also detected, including 31 methyl and 17 ethanalyl sites. Accompanying oxidations were frequent but not counted.

All peptides detected and found as modified in the database search were numbered, and then areas under the peak were quantified manually (Appendix A). Of those, 157 were used in the statistical analysis.

Interestingly, several amino acid sequences containing different modification patterns were found and quantified more than once. For example, one peptide from angiotensinogen (35 and 179) was found with kynurenine modifications, accompanying oxidations and formylation (Figure 2A). The amino acid sequence of peptides 1, 164 and 165, stemming from transthyretin, was found and quantified three times separately, carrying multiple combinations and forms of modifications with different abundances. In all three forms, an oxidative modification, kynurenine or hydroxyl-kynurenine, was detected (Figure 2B).

### 3.3. Modified Peptides Are Differently Abundant in CSF from AD and OND Patients

In sum, quantitation of the area under the peak of high-quality identified peptides resulted in 113 candidate peptides, which could be considered for their discriminating power of AD and OND patients (Appendix A). The area under the receiver operating characteristic curve (ROC-AUC) analysis was used to select the most promising candidate peptides. However, statistical significance cannot be achieved with the sample numbers of this study. It revealed that 17 peptides possessed a value of 0.8 or higher (Table 2). In general, most of the candidate peptides and all 17 high ROC-AUC candidates were more abundant in the AD samples, indicating an accumulation of all types of modifications.

Interestingly, four peptides from the top 17 candidates contain early glycative modifications, which were not as frequently detected. Furthermore, eight peptides presented at least one stable glycoxidative or oxidative modification, often accompanied by oxidations on various amino acids. In contrast, eight potential AGEs were found on the candidate peptides, which is less than expected. These observations hint at a specific diversification of modifications compared to overall counts, indicating a potential for transient and specific sugar- and oxidation-related changes.

In general, the 17 peptides presented in Table 1 would be the most promising biomarker candidates for validation in a larger cohort since they show high potential discrimination between AD and OND.

### 3.4. Peptide Candidates Are Not Enriched but Correlate with Diagnostic Measures and the Age of Individual Patients

The combined analysis of the AU curves of the candidate peptides, including correction for multiple testing (HAUCA), did not reveal a specific enrichment of candidates with high ROC-AUC values as compared to the random expectations (Figure 3A, blue vs. black curve).

Subsequently, Spearman’s rank correlation analysis showed that most of the candidate peptides showed a strong or very strong positive correlation with the age of the patients (12 peptides). Also, there is at least a moderate correlation with some of the established biochemical core biomarkers (Figure 3B). Among the candidates, a uniform and mostly positive moderate to very strong correlation is observed with t- and p-tau, with median correlations of 0.43 in all candidates and 0.58 and 0.51 in peptides with a ROC-AUC ≥ 0.7. Of the top 17 candidates, 114 and 151 very strongly (<0.8) correlate with t-tau and 56, 134, 2_166 and 31 very strongly correlate with both t-tau and p-tau. Furthermore, eight additional peptides strongly (>0.6) correlate with t-tau, and eight strongly correlate with p-tau.

In contrast, the median correlation with single amyloid-beta measures is more diverse and very weak in Aβ 1-42 (−0.07, 0.06 for ≥0.7), and no peptide from the top candidates shows a very strong correlation. In contrast, correlations to the Aβ ratio are primarily weak to moderate and negative (median −0.32, −0.45 for ≥0.7) and more uniform. However, from the top candidates, peptides 125, 35 and 90 show a very strong negative correlation with the Aβ-ratio, and nine peptides are strongly correlated.

Of note, in our study, the age of the patients also correlated very strongly with t-tau (0.88) and strongly with the Aβ-ratio (−0.78) and p-tau (0.88) (see Appendix A and Appendix A). Furthermore, peptides with AUCs >0.7 and only very weak to weak correlation with age correlate less strongly with the Aβ-ratio and tau markers but, on average, more strongly with Aβ 1-42 (Appendix A). Two exemplary peptides for that finding are 150 and 131.

Peptide 203, a kyn-modified peptide of alpha-1-acid glycoprotein (ROC-AUC 0.8), showed correlations not resembling the high ROC-AUC candidates, e.g., negative correlations to Aβ 1–40 and 42 and p-tau.

## 4. Discussion

The FDA-NIH Biomarker Working Group defines a biomarker as an indicator of biological or pathological processes or a response to an intervention or exposure [45]. AGEs and oxidative protein modifications can be formed due to underlying biological and pathological mechanisms relevant to AD. Altered glucose metabolism in AD brains can explain the induced formation of glycative modifications and AGEs, while the evoked oxidative stress might lead to oxidative protein modifications [46,47]. In concordance with that idea, most modified peptides were more abundant in AD than in the control samples.

Among the candidate peptides with high ROC-AUCs, some stemmed from proteins already described to convey physiological and pathological mechanisms in the brain. For example, peptides 35 and 179 contain the amino acid sequence of an unordered region in angiotensinogen, the angiotensin precursor protein. Angiotensin and cleaving proteases are regularly reported to be modulated in AD, albeit with contradictory magnitudes and results [48]. Increased oxidative and glycoxidative modifications on the precursor angiotensinogen could, on the one hand, influence either cleavage or binding of the proteases to the protein and, on the other hand, mirror the combination of increased abundances and a reactive environment in the brain/CSF and would be worth further investigation.

In general, the presence of glycative and glycoxidative modifications of different origins and in early, intermediate and stable late stages hints towards the presence of glucose, fructose, ribose and several dicarbonyls (GO, MGO, 3-DG) in CSF or neuronal cells. Also, it indicates changes between the groups that could have happened in different time frames from days to months. The same holds for oxidative modifications.

Furthermore, this study, for the first time, reported the detection of kynurenine, hydroxyl-kynurenine and oxolactone as part of peptides in CSF. In general, free kynurenine and hydroxyl-kynurenine emerge from a mechanism different from protein-bound ones. Free kynurenine and hydroxyl-kynurenine, among others, are formed enzymatically through a multi-enzyme cascade from tryptophan [49]. In contrast, peptide-bound modifications report non-enzymatic formation in specific environments and, thus, mirror the systemic oxidative stress rather than tryptophan catabolism [50], proving the feasibility of investigating these modifications as proxies for oxidative stress [36]. The same reporting function for the oxidative environment already described to persist in AD was also observed in the detection and quantification of amide AGEs, which are also formed under oxidizing conditions [51]. From this point of view, modified peptide biomarkers with glycative, glycoxidative or oxidative modifications could be best suited to monitor the change or persistence of metabolic changes and oxidative conditions in the brain or at least CSF throughout the AD continuum similar to the clinical use of HbA1c in diabetes.

However, to enable clinical use of modified peptide markers, the findings would first need validation and further description of the occurrence of the peptides. Additionally, a precise, either MS or antibody-based detection method is needed to monitor specific modified peptides with a defined set of modification possibilities. This might be challenging with the current discovery-type workflow. As for favorable candidates, the identified peptide belonging to angiotensinogen would be of interest for this kind of validation because of its localization on the protein in an easily accessible unordered region and the knowledge generated about possible modifications from this study.

Our study analyzed peptide-bound modifications by mass spectrometry in contrast to previous studies where free AGEs or tryptophan metabolites like kynurenine were investigated. The results from those studies were often contradictory, e.g., for CML, where different studies demonstrated either an increase or no change in AD, possibly dependent on the detection method used [30,31,32]. In general, antibody-based AGE detection faces several limitations, including low epitope specificity and inhibition by autoantibodies [52], and, thus, can be inferior to MS analyses. On the other hand, HPLC or MS detection of acidic hydrolyzed samples is subject to artifact generation due to harsh conditions. In contrast, the peptide-based analysis by MS is preceded by a mild sample preparation. While the preparation and analysis can also be challenging, the results can provide different layers of additional information, e.g., the identity of the modified protein, a certain site specificity and the co-existence of different modifications [53]. Here, the untargeted approach also poses additional benefits compared to MRM-MS/MS. As we could observe in this study, peptides comprising identical amino acid sequences but different modifications or modification patterns showed distinct abundances. This information can only be harnessed in a non-targeted, peptide-based analysis and might be crucial for selecting peptides as biomarkers.

An untargeted MS approach is currently the most promising in resolving glycative modifications. Many of those are prone to react further and, thus, create different modification patterns depending on the sample preparation and peptide environment, e.g., pH or oxidative substances. For example, the detection of CE or CM moieties can result from labile hydromididazolones fragmenting during mass spectrometry analysis [54]. According to that, CM and CE site abundances are expected to be overestimated. However, our study also showed that detecting more transient and intermediate glycative modifications is possible by detecting fructosylated, glyoxal-imine or hydroimidazolone sites.

For decades, high-resolution mass spectrometry with high-accuracy hybrid instruments, like QqTOF-, Q-Orbitrap- and LIT-Orbitrap-MS, has been recognized as the method of choice in nanoLC-based bottom-up-proteomics studies in general and Maillard reaction product proteomics in particular [55,56]. The quality of the resulting MS/MS spectra is essential for identification, precise localization of the glycated site and modification identity. Our previous studies indicated that softer collision conditions established in the HR mass analyzers are associated with higher reliability of peptide identification and yield high identification rates for both early and advanced glycated peptides and proteins [57,58,59]. Thus, LIT-Orbitrap-MS is preferable for the analysis of modified peptides.

Exploratory studies, especially in the field of omics, often suffer from a lack of statistical power due to small cohort sizes and large numbers of features. The same holds for this feasibility study, which is also limited. Thus, we could not report statistical significance or enrichment for peptide candidates in the HAUCA analysis.

Nevertheless, the analysis of correlations between diagnostic biochemical core biomarkers and peptides hinted towards underlying mechanisms being reflected in the candidates. From the top candidates, it becomes clear that some peptides most strongly correlate with t-tau and p-tau, linking them to the tau pathology, while others most strongly correlate with the Aβ ratio. The second group of peptides might be interesting as putative biomarkers since the ratio was already shown to correlate negatively with cognitive decline and Aβ-PET scans independent of diagnosis [60]. It would be interesting to investigate whether specific types of modifications are associated with those different pathologies in a larger cohort.

One peptide from alpha-a1-glycoprotein modified with kynurenine (203) did not correlate strongly with age and showed distinctly different correlations from the other top-candidate peptides. This might be attributable to the proteins’ role as acute-phase protein response to inflammation. At the same time, it is an example that oxidative stress can occur independently since kynurenine and other oxidative modifications were found on a range of peptides with different correlations, e.g., 151 with very high correlations to t-tau, 35 with strong correlations to the Aβ ratio and 203 mentioned before. Thus, our data support the general notion of monitoring oxidative stress in patients with risk factors for neurodegenerative diseases by established methods like PET [61] and novel methods [62].

However, most of the top-ranking candidates also strongly correlate with the age of the patients. The correlation results might be confounded since age differs between the AD and OND groups (Table 1). Separating the ROC-AUC < 0.7 candidate peptides with weak and stronger correlations to age, a change in the overall correlations to core biomarkers could be observed. Thus, age seems to have a confounding influence on some candidates. In this regard, our study matches current results, where age-related effects on cognition were primarily mediated by tau-deposition even outside of the AD continuum [63], since we also observed high correlations with t-tau for the candidate peptides and with age. Furthermore, sex and apolipoprotein E (APOE) status might be other confounding factors already observed in other AD biomarkers [64]. In general, we contributed to resolving the question of presence, detectability and quantitative analysis of glycative, glycoxidative and oxidative modifications on peptides. Nevertheless, due to the small sample size, this study cannot conclude the usability of the determined candidate peptides as biomarkers.

In any case, future research is warranted to establish glycatively and oxidatively modified peptides as early biomarkers of AD. At first, the presented approach should be used in a larger cohort, including additional groups like patients at risk without cognitive impairment, MCI and different stages of AD, to investigate the presence and magnitude of peptide-bound modifications along the AD continuum. To validate the candidates as diagnostic markers, it would be necessary to determine the discrimination power against other types of dementia or mixed pathologies. From the current knowledge, it is more likely that the markers will help identify persons with a high risk of developing neurodegenerative diseases in general. Especially for peptides detected in multiple modification states, it would be interesting to monitor those across the AD continuum and investigate a putative change evoked by the intensifying pathogenic mechanisms.

## 5. Conclusions

In this feasibility study, we set out to prove the detectability of endogenous glycatively and oxidatively modified peptides in CSF by mass spectrometry, which we achieved with 299 sites on 219 peptides for a wide variety of modifications from different precursor molecules and at different stages of reaction without any prior enrichment.

Additionally, we could point out the advantages of peptide-based HR-MS analysis of AGEs and related modifications compared to the analysis of free AGEs and tryptophan metabolites in patient samples. Of particular interest might be tryptophan side-chain oxidations to monitor oxidative stress and early glycative modifications from fructose, glucose and ribose as well as intermediates and amide-AGEs.

Furthermore, we showed that it is possible to quantify specific changes in modified peptides between the AD and OND groups in a large group of peptides (157) from as few as five samples per group. Anyhow, because of the limited sample size, this pilot study cannot prove that specific peptides are biomarkers for AD, whether diagnostic or prognostic. In that regard, the results are preliminary, and validating the candidate peptides might not be possible. However, we could demonstrate that the change in abundance of the peptides correlates with the change in biochemical core biomarkers Aβ, p-tau and t-tau as well as with the age of the patients and that the strength of the correlations depends on the correlation with age.

These findings point towards the usability of AGE or oxidatively modified peptides as biomarkers, possibly in monitoring metabolic changes and oxidative stress. Furthermore, the results revealed 17 peptide candidates with high ROC-AUC values > 0.8. However, the findings of this pilot study need to be validated in an extended study with more individuals, preferably over the whole AD continuum and with other pathologies. This step would be of utmost importance due to the methodological limitations of this study to establish modified peptides as new diagnostic biomarkers for AD or as prognostic/staging markers for neurodegenerative diseases from our dataset.

## Figures and Tables

**Figure 1 biomedicines-12-02127-f001:**
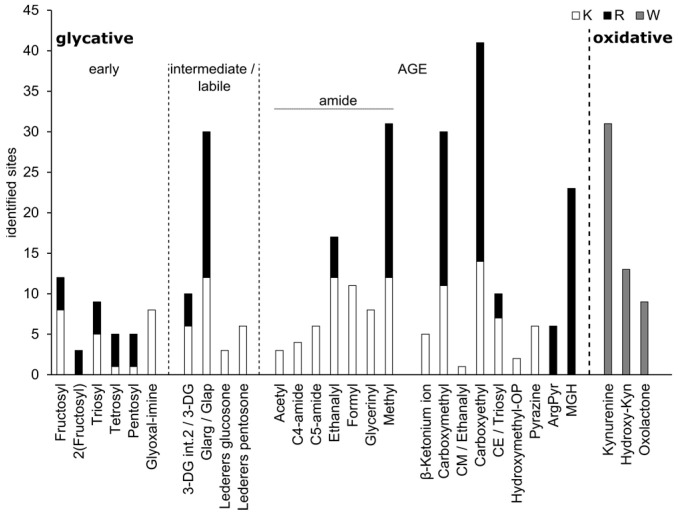
Count of sites containing glycative modifications (early and intermediate), advanced glycation endproducts (AGEs) and oxidative modifications identified in cerebrospinal fluid (CSF) samples of AD patients and OND controls. Glycative modifications comprise early modifications, where monosaccharides reacted with amino acids, forming Schiff bases of trioses, tetroses, pentoses and fructose (hexose) as well as the carbonyl glyoxal (glyoxal-imine). Intermediate modifications include carbonyl- and glucose-derived labile structures, which can react further to imidazolones, pyrraline (3-DG), CM-AGEs (Glarg/Glap), glucosepane (glucosone) or pentosidine (3-DG and pentosone). AGEs, which are stable glycative end products, include classical modifications; and the novel group of amide-AGEs, which include glycoxidation products of Amadori intermediates. Oxidative modifications are stable products of tryptophan residue oxidation. Database search results are summed counts for each specific modification. Counts were combined in one column if modifications were present at lysine (K) and arginine (R). “/” denotes sites where a specific modification was not identified, but two possibilities were detected. K: lysine, R: arginine, W: tryptophan, 3-DG: 3-deoxyglucosone, int.2.: intermediate 2, ArgPyr: ArgPyrimide, MGH: methylglyoxal-hydro imidazoline.

**Figure 2 biomedicines-12-02127-f002:**
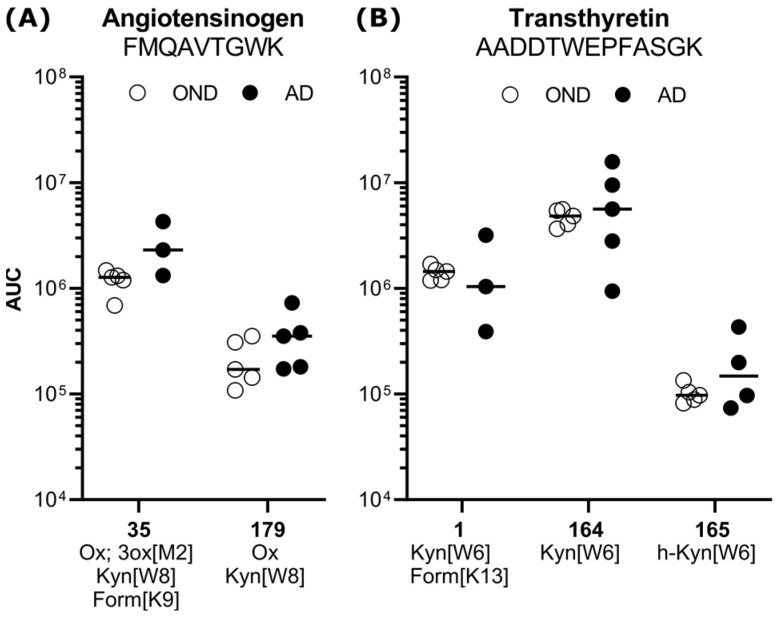
Exemplary quantification results of modification-specific forms of peptides in the OND control group (white circles) vs. AD, black circles with mean (dash). (**A**) Peptide FMQAVTGWK from angiotensinogen contains kynurenine (179) and a formyl modification (35). (**B**) Peptide AADDTWEPFASGK from transthyretin was identified and quantified with various oxidative tryptophan modifications and a glycoxidative formyl group. White circles represent the OND, black circles the AD group. Ox: oxidation, 3ox: trioxidation, Kyn: kynurenine, Form: formyl, h-Kyn: hydroxyl-kynurenine, AUC: area under the peak curve.

**Figure 3 biomedicines-12-02127-f003:**
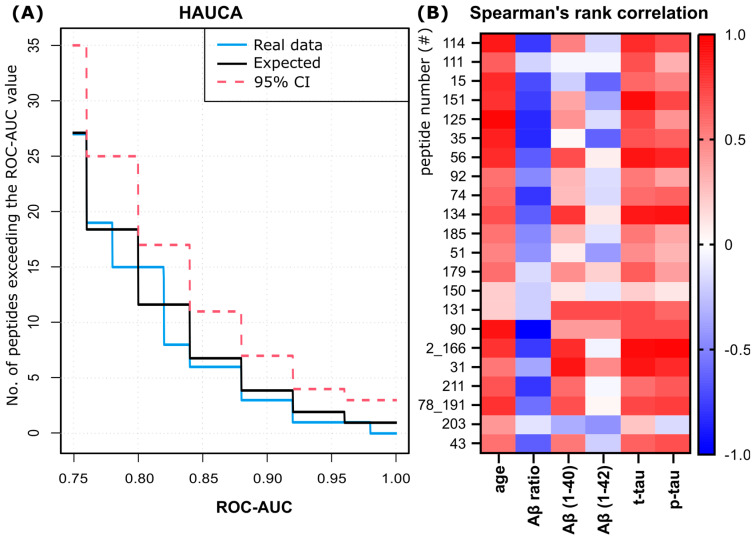
Statistical analysis of peptide candidates. (**A**) The HAUCA analysis curve for candidate peptides indicates expected results (black curve), observed data (blue curve) and 95% confidence interval (CI) for enrichment of peptides (red dashed line). HAUCA analysis can be used to evaluate small datasets for enrichment of biomarker candidates, in which case the data curve should deviate from the black curve. (**B**) Spearman’s rank correlation coefficients of single peptide signal values with age and biochemical core biomarkers. Correlations can assume values between −1 and 1. A negative correlation is shown in blue, and a positive correlation is shown in red. Peptide number (#) refers to the numbers in column “No (#)” in Appendix A.

**Table 1 biomedicines-12-02127-t001:** Patient characteristics.

	Control (OND) (*n* = 5)	AD (*n* = 5)	Diagnosis Cut-Off AD *	*p*-Value
Median age	63 (55–74)	80 (78–85)		0.0009
Gender (m/f)	5/0	2/3		
MMSE score	29	13		0.0000001
Aβ_40_ (pg/mL)	11,547	12,075		0.870
Aβ_42_ (pg/mL)	854	511.4	>551	0.129
Aβ-ratio (Aβ_42_ × 10/Aβ_40_)	0.784	0.442	<0.46	0.046
p-tau (pg/mL)	52.24	97.82	>61	0.082
t-tau (pg/mL)	296.8	961.4	<293	0.081

OND: other neurological diseases, AD: diagnosed Alzheimer’s disease, *p*-value: statistical significance value for the differences between OND and AD determined by unpaired *t*-test, MMSE: mini mental state examination, Aβ_40_: amyloid beta 1–40, Aβ_42_: amyloid-beta 1–42, p-tau: phosphorylated tau-protein at threonine 181, t-tau: total tau-protein, * cut-off values for diagnosis as reported in the manuals of the kit vendors/by laboratory.

**Table 2 biomedicines-12-02127-t002:** Candidate peptides with a potential of discriminating AD from ONDs and a receiver operating curve—area under the curve(ROC-AUC) > 0.8.

#	Peptide Sequence	Modification Sites	Master Protein ID	Master Protein Name	ROC-AUC	Cor.
114	R_[pent]_GR_[pent]_LKNAGGGIR_[cm]_	R_23_, R_25_, R_34_	Q8NAT4	cDNA FLJ34815 fis, clone NT2NE2007786	1	+
111	QWW_[oxo]_R_[glarg]_PWVDHASSR	W_97_, R_98_	V9GZ18	Piezo-type mechanosensitive ion channel component 2	1	+
15	C_[cam]_VLPPMDGYPHC_[cam]_EGK_[gi]_	C_145_, C_156_, K_159_	Q09328	Alpha-1,6-mannosylglycoprotein 6-beta-N- acetylglucosaminyltransferase A	1	+
151	_[kyn *]_WWFDWGWC_[cam]_NIR_[etha]_	W_26/27_, C_33_, R_36_	C6GLZ9	Uncharacterized protein	1	+
125	SSIPRPR_[fruc]_SWALGR_[glarg]_	R_98_, R_104_	L8E8L4	Alternative protein CEP164	1	+
35	_[ox *]_FM _[3ox]_QAVTGW_[kyn]_K_[form]_	M_261_, W_267_; K_268_	Q59EP2	Angiotensinogen	0.93	+
56	KC_[cam]_R_[cm]_PIIC_[cam]_DK_[glap]_YC_[cam]_PLGLLK_[glap]_	C_539_, R_540_, C_544_, K_546_, C_548_, K_554_	Q9NZV1	Cysteine-rich motor neuron 1 protein	0.88	+
92	NIFPIW_[ox]_ALGR_[ce/trio]_	W_714_, R_718_	B2R694	Terpene cyclase/mutase family member	0.88	+
74	LR_[ce]_LSLRNMPVVPC_[cam]_	R_51_, C_62_	A0A087X1L6	Eukaryotic translation initiation factor 2D	0.85	+
134	TQNLEQK_[trio]_LSGDSR_[trio]_AC_[cam]_R_[mgh]_	K_112_, R_118_, C_120_, R_121_	B3KRU0	cDNA FLJ34916 fis, clone NT2RP7001475, highly similar to Ankyrin repeat domain-containing protein 6	0.85	+
185	KLSSW_[kyn]_VLLM_[ox]_K	W_262_, M_266_	A0A024R6I7	Alpha-1-antitrypsin	0.85	+
51	ITFSDVR_[mgh]_PNQQEY_[ox]_K_[glyc]_ISSF_[ox]_EQR	R_897_, Y_903_, K_904_, F_908_	Q86TC9	Myopalladin	0.84	+
179	_[ox *]_FMQAVTGW_[kyn]_K	W_267_	Q59EP2	Angiotensinogen	0.84	+
150	WR_[glarg]_NSEIQC_[cam]_YVNGQLVSYGDMAW_[kyn]_HVNTNDSYDK_[form]_	W_174_, C_180_, W_194_, K	D3DP68	Chromosome 2 open reading frame 27, isoform CRA_a	0.833	+
131	TK_[h-op/ce]_EKYIDQEELNKTKPIW_[h-kyn]_TR	K_280_, W_296_	Q14568	Heat shock protein HSP 90-alpha A2	0.833	+
90	_[ox *]_NEGFF_[2ox]_ALC_[cam]_K_[3dg-i2]_GF_[2ox]_WPNW_[3ox]_LR	F_259_, C_262_, K_263_, F_265_, W_269_	B3KSR0	Kidney mitochondrial carrier protein 1	0.833	+
2_166	AAFALGGLGSGFASNR_[ce/trio]_	R_571_	P27708	Multifunctional protein CAD	0.8125	+

cor. = correlation, + = positive with AD, pent: pentosyl, cm: carboxymethyl-adduct, gi: glyoxal-imine, cam: carbamidomethyl-adduct, glarg: glyoxal-derived dihydroxy-imidazolidine, ox: oxidation, 2ox: dioxidation, 3ox: trioxidation, kyn: kynurenine, form: formyl-adduct, glap: glyceraldehyde-derived pyridinium, ce: carboxyethyl-adduct, trio: triosyl, oxo: oxolactone, pyr: pyralline, fruc: fructosyl, h-op: hydroxymethylOP, h-kyn: hydroxy-kynurenine, 3dg-i2: 3-deoxyglucosone intermediate 2, etha: ethanalyl, pyra: pyrazine, mgh: methylgyoxal-hydoimidazolone, glyc: glycerinyl, _*_ = modification site ambiguous.

## Data Availability

A data summary is contained within the Appendix A. The data that support the findings of this study are available from the corresponding author upon reasonable request.

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
