# Peer review of "Peptide-Bound Glycative, AGE and Oxidative Modifications as Biomarkers for the Diagnosis of Alzheimer’s Disease—A Feasibility Study"

_biomedicines, 2024, doi:10.3390/biomedicines12092127_

Round 1

Reviewer 1 Report

Comments and Suggestions for Authors

The manuscript presented by Anne Grosskopf and co-workers. title: “Peptide-bound glycative, AGE- and oxidative modifications as biomarkers for the diagnosis of Alzheimer’s disease – a feasibility study” is well written, clear, and easy to read. The topic is interesting and therefore, it adds new and interesting information to the subject area of the type the diagnosis of Alzheimer’s disease (in this case peptide-glycated by AGE, due to diet rich of sugar and fat) and the influence of it on burden condition of Alzheimer’s disease an explored scientific area for possible biomarkers.

Said that, it is useful to add more details on other neurological diseases since the “limitation” to use the CFS.

Minor

line 92 please avoid the use of Alzheimer’s dementia

Author Response

We thank the reviewer for the appreciation and interest in the research topic.

Said that, it is useful to add more details on other neurological diseases since the “limitation” to use the CFS.

We added additional information in the introduction (l. 46-71) regarding the limitation of CSF and also we elaborated on other biomarker groups in AD and included the notion of specific or unspecific nature of different biomarker groups regarding AD.

Minor

line 92 please avoid the use of Alzheimer’s dementia

We agree with the reviewer on the wording replaced the term Alzheimer’s dementia.

Reviewer 2 Report

Comments and Suggestions for Authors

Review of the Article: "Peptide-bound Glycative, AGE- and Oxidative Modifications as Biomarkers for the Diagnosis of Alzheimer’s Disease – A Feasibility Study"

Recommendations for Authors

1.      The introduction should include more comprehensive background information on the current state of biomarker research for Alzheimer’s disease (AD), specifically discussing existing CSF biomarkers beyond amyloid beta and tau proteins. Additionally, providing a clearer rationale for focusing on glycation, AGE, and oxidative modifications as potential early biomarkers would strengthen the study's premise.

2.      The study's objectives are not explicitly stated in the introduction. It would be beneficial to clearly outline the specific goals, such as identifying peptide-bound modifications in CSF and their potential as early biomarkers for AD.

3.      The criteria for selecting AD patients and controls with other neurological diseases (OND) should be clearly described. This includes information on how patients were diagnosed and what specific characteristics were required for inclusion in the study.

4.      The manuscript mentions the control group having other neurological diseases but lacks detail on the types of diseases included. A table or additional text explaining these conditions and their relevance or lack of overlap with AD biomarkers would add clarity.

5.      The methodology, particularly the CSF sample collection, processing, and storage procedures, should be described in more detail to ensure reproducibility. Information on how samples were handled to prevent degradation or modification should be included.

6.      It would be helpful to explain why specific peptide modifications (glycation, AGE, and oxidative) were targeted in this study. Briefly review their biological significance and potential relevance to AD pathology.

7.      The statistical methods used to evaluate the data, such as the receiver operating characteristic (ROC) curves and the handling of missing data, should be explained in more detail. Given the small sample size, this includes justification for using ROC-AUC values and the statistical power considerations.

8.      Provide a rationale for the chosen sample size of 10 patients. Discuss how this number was determined and whether it was based on previous studies, statistical power calculations, or practical considerations.

9.      The study's limitations should be more comprehensively discussed, including the small sample size, potential confounding factors, and the preliminary nature of the findings. Acknowledging these limitations openly will help contextualize the results and their applicability.

10.   The results section could benefit from more detailed descriptions and interpretations of key findings. For example, explain the significance of the identified peptide modifications and their potential as biomarkers more clearly.

11.   Ensure all figures and tables are clearly labeled and include comprehensive legends. For example, Figure 1 would benefit from a more detailed explanation of the types of modifications and their relevance. Consider adding a visual summary or flowchart of the study design to assist readers in understanding the workflow.

12.   The discussion should more thoroughly explore the clinical implications of the findings. This includes how these peptide modifications could be used in early diagnosis or monitoring of AD and what further steps are needed to translate these findings into clinical practice.

13.   The conclusion should succinctly summarize the main findings and their significance, clearly stating whether the study supports the feasibility of using these peptide modifications as biomarkers. Highlight the need for larger, more comprehensive studies to validate these results.

14.   Some references cited in the manuscript are older and might not represent the most recent advancements in the field. Updating the reference list to include more recent publications, especially those within the last five years, will enhance the manuscript’s relevance.

15.   Besides the limitations, the discussion should include suggestions for future research. This could involve validating findings in a larger cohort, exploring other potential biomarkers, or conducting longitudinal studies to assess changes over time.

16.   The choice of nanoLC-LIT-Orbitrap-MS/MS for peptide analysis should be justified by discussing its advantages over other techniques in detecting modifications relevant to AD.

17.   Provide a more detailed interpretation of the correlations between peptide modifications and established biochemical markers. Discuss what these correlations might imply about the underlying disease mechanisms.

Comments on the Quality of English Language

The manuscript is generally well-written, but some sentences could be rephrased for clarity and flow. A minor round of editing to correct grammatical errors and improve sentence structure would enhance readability.

Author Response

Review of the Article: "Peptide-bound Glycative, AGE- and Oxidative Modifications as Biomarkers for the Diagnosis of Alzheimer’s Disease – A Feasibility Study" - Recommendations for Authors

Reviewer 2:

We thank the reviewer for the comments and greatly appreciate the effort to point out improvements to the paper in a detailed fashion. We tried to improve the manuscript accordingly and will give a point-to-point reply for the changes we made when possible.

  1. The introduction should include more comprehensive background information on the current state of biomarker research for Alzheimer’s disease (AD), specifically discussing existing CSF biomarkers beyond amyloid beta and tau proteins. Additionally, providing a clearer rationale for focusing on glycation, AGE, and oxidative modifications as potential early biomarkers would strengthen the study's premise.

We thank the reviewer for the suggestion, we added a paragraph (p2, l 50-62) about additional biomarker-groups which reads as follows:

Apart from the core amyloid (A) and tau (T) fluid biomarkers for diagnostics, addi-tional markers are currently investigated and considered, especially for staging, prognosis and identification of co-pathologies [1].

Markers of neuronal injury, dysfunction and degeneration, summarized as N-group, include neurofilament light chain (NEFL), Visinin-like protein-1 (VILIP-1), synaptoso-mal-associated protein-25 (SNAP-25) and neurogranin [7,8]. NEFL levels, e.g., reflect white-matter axonal damage in various neurological diseases and dementias and are highly clinically relevant for staging and progression [9].

Another group of emerging protein markers is related to brain inflammation (I) me-diated by astrocytes and microglia. It includes interferons, soluble TREM2 (sTREM2) and glial fibrillary acidic protein (GFAP), as well as chitinase-like 3 protein 1 (YKL40) [1,7]. GFAP and YKL40 are involved in astrogliosis, mediating early disease progression [10]. Apart from the mentioned groups, additional markers for vascular damage (V) or synu-cleinopathy (S) are established to provide information on relevant co-pathologies [1].

Furthermore, we tried to clarify the rationale for the choice of glycative and oxidative modifications by adding more background information on page 2, l 66-74:

Preceding inflammation and neuronal injury, increased oxidative stress by impaired mitochondrial function is described in aging and neurodegenerative diseases like AD  [11]. Furthermore, it mechanistically links many described risk factors of AD like cardio-vascular diseases (CVD), insulin resistance, obesity and type-2 diabetes mellitus (T2DM) [12-15]. Furthermore, it was established that a deregulated glucose metabolism in inherent for both early AD and T2DM. Indeed, recent proteomics approaches to uncover early biomarkers of AF in CSF revealed prominent changes in enzymes of the glucose metabolism, but there are no established biomarkers, yet [16-19].

It is feasible to assume that damage to proteins induced by reactive oxygen species (ROS) or sugar molecules via the Maillard reaction could be detected even before changes in protein abundances similar to HBA1C in the onset and monitoring of diabetes.

  1. The study's objectives are not explicitly stated in the introduction. It would be beneficial to clearly outline the specific goals, such as identifying peptide-bound modifications in CSF and their potential as early biomarkers for AD.

We agree with the reviewer and re-wrote the last paragraph of the introduction to more concisely report our ojectives. It reads as follows:

In this feasibility study, we aim to identify and quantify endogenous, stable glycated, AGE- and oxidatively modified peptides from CSF-proteins of AD patients compared to a control group without prior enrichment. Additionally, we intend to determine their potential as early biomarkers for diagnosing AD.

  1. 3.      The criteria for selecting AD patients and controls with other neurological diseases (OND) should be clearly described. This includes information on how patients were diagnosed and what specific characteristics were required for inclusion in the study.

Unfortunately, detailed information on how the diagnosis for each patient was made was not available due to data-protection reasons.

  1. The manuscript mentions the control group having other neurological diseases but lacks detail on the types of diseases included. A table or additional text explaining these conditions and their relevance or lack of overlap with AD biomarkers would add clarity.

For points 3 and 4, we added a paragraph in the methods section in chapter 2.1. Unfortunately, detailed information on how the diagnosis was made was not available. The additions in the manuscript reads as follows:

Other neurological diseases permitted as controls were pain disorders, schizophrenia, depressive disorders and delirium. Patients with other types or mixed forms of dementia and with evident cognitive impairment (MMSE < 24) were excluded from the control group to minimize the overlap with AD pathology and biomarkers in this feasibility study. For the AD group, physician-diagnosed AD was a prerequisite, as was cognitive impairment (MMSE < 24). Furthermore, patients with mixed pathologies were excluded.

  1. The methodology, particularly the CSF sample collection, processing, and storage procedures, should be described in more detail to ensure reproducibility. Information on how samples were handled to prevent degradation or modification should be included.

We added to pieces of text in the manuscript, to provide additional information on CSF collection, storage and sample handling (p3, l109-118):

CSF was obtained by lumbar puncture as follows. The patient’s back was sterilized, covered with a sterile cloth with a hole and sterilized again. At first, an introducer was applied to the puncture site. Subsequently, the spinal canal was punctured horizontally with a slight cranial direction using an atraumatic 22G LP-needle type Sprotte. After removal of the stylet, CSF was collected by a qualified study nurse in six numbered 10 mL polypropylene screw cap tubes. The first three tubes were used for diagnostics.  CFS samples (2 ml each) in the other three tubes were stored in an ice-cooled box and stored at -80°C immediately after the end of the procedure until further use for research purposes without centrifugation.

and additionally (l. 129-131)

After thawing on ice, CSF samples were aliquoted in 1.5 ml polypropylene tubes and directly subjected to protein determination via Pierce 660 nm Protein Assay (#22660, Thermo Fisher Scientific) and then enzymatic digestion.

  1. It would be helpful to explain why specific peptide modifications (glycation, AGE, and oxidative) were targeted in this study. Briefly review their biological significance and potential relevance to AD pathology.

We thank the reviewer for that suggestion. We tried to add explanations and information to the biological significance and relevance to AD throughout the introduction.

  1. 7.      The statistical methods used to evaluate the data, such as the receiver operating characteristic (ROC) curves and the handling of missing data, should be explained in more detail. Given the small sample size, this includes justification for using ROC-AUC values and the statistical power considerations.

We included additional references as well as information in the section 2.9. It now reads (changes highlighted):

Peptides with the derived AUCs were considered for further analysis if the corresponding signals were present in at least three of five replicate samples of both groups. Furthermore, peptides with two signals in one group but more than three in the second were also considered. Missing data was not imputed unless described otherwise. The area under the receiver operating curve (ROC-AUC) was calculated for each peptide as described previously based on the selection of the 20 best attributes for classifier training and a leave-one-out (jackknife) cross-validation strategy [45,46]. The results were utilized to es-timate the potential of the candidate peptides as prospective disease-related biomarkers for Alzheimer’s (AD) versus other neurological diseases (OND). Due to the small cohort size and the large number of peptides considered, after any correction for multiple testing, even a perfect ROC-AUC value of 1 would no longer have statistical significance. Thus, no correction for multiple testing was carried out in this analysis. However, even without statistical significance, AUC values can help determine the candidate peptides with the best potential to serve as biomarker candidates. One way to evaluate whether a small data set contains potential biomarkers is by calculating the high abundance AUC analysis (HAUCA) [46].To generate the high HAUCA curves to evaluate the enrichment of dis-criminating peptides, missing values were substituted by the median value of the respective groups (OND or AD) and Bonferroni-Holm correction for multiple testing was applied [46,47].

Additionally, Spearman’s rank correlations were calculated for all peptide candidates and the respective patients’ diagnostic biochemical parameters (see Table 1) and age.

All statistical analyses were carried out with R software version 4.2.2. The R packages used were “pROC” by Robin and colleagues for ROCS-AUCs [48] and “Hmisc” for the imputation of missing values. The HAUCA analysis was recently included in the R shiny app “HiPerMAb” [45].

  1. Provide a rationale for the chosen sample size of 10 patients. Discuss how this number was determined and whether it was based on previous studies, statistical power calculations, or practical considerations.

We thank the reviewer for this question. Of course, we know that our sample size is small, and the results are thus debatable. However, at the time of the feasibility study, it was not possible to analyze more than the ten samples described here via MS until the end of the project funding due to time and machine capacity limitations.

  1. The study's limitations should be more comprehensively discussed, including the small sample size, potential confounding factors, and the preliminary nature of the findings. Acknowledging these limitations openly will help contextualize the results and their applicability.

To accommodate the reviewer’s suggestions, we included a paragraph regarding confounding factors, especially in the correlation analysis, which reads:

However, most of the top-ranking candidates also strongly correlate with the age of the patients. Since age differs between the AD and OND groups (Table 1), the correlation results might be confounded. Separating the ROC-AUC <0.7 candidate peptides with weak and stronger correlations to age, a change in the overall correlations to core biomarkers could be seen. Thus, age seems to have a confounding influence on some candidates. In this regard, our study matches current results, where age-related effects on cognition were primarily mediated by tau-deposition even outside of the AD continuum {Wuestefeld, 2023 #538}, since we also observed high correlations with t-tau for the candidate peptides and with age. Furthermore, sex and AOPE status might be other confounding factors already seen in other AD biomarkers {Milà-Alomà, 2021 #506}.

In general, we contributed to resolving the question of presence, detectability and quantitative analysis of glycative, glycoxidative and oxidative modifications on peptides. Nevertheless, this study cannot conclude the usability of the determined candidate peptides as biomarkers due to the small sample size.

See also point 12 for additional discussion on limitations.

We also included a small passage in the conclusion regarding the limitations:

Anyhow, because of the limited sample size, this pilot study cannot prove that specific peptides are biomarkers for AD, whether diagnostic or prognostic. In that regard, the results are preliminary, and validating the candidate peptides might not be possible.

  1. 10.   The results section could benefit from more detailed descriptions and interpretations of key findings. For example, explain the significance of the identified peptide modifications and their potential as biomarkers more clearly.

We agree with the reviewer and explained the results in more detail throughout the results section.

For Identification of modified peptides, we added (l. 273-277, l. 280-282, l. 284-286):

However, the data provides evidence of transient reaction of trioses, tetroses, pentoses, presumably ribose, and fructose, as well as glyoxal (GO) with CSF proteins as early glycative modifications. This indicates the presence and change in abundance of those sugars and carbonyls, probably through changing metabolic pathways like glycolysis or the pentose phosphate pathway.

3-deoxyglucosone intermediates (3-DG) are fructose-derived precursors of pyralline, while Lederers glucosone and pentosone, which were also detected, can react to glucosepane and pentosidine, respectively.

Apart from pyrazine, detected only 6 times and derived from GA, all other stable AGEs are end products of GO and MGO reactions. Apart from those, many different glycoxidative amide AGEs formed from sugars only in oxidative conditions were also detected, including 31 methyl- and 17-ethanalyl-sites.

For the quantification and description of the top 17 candidates:

Interestingly, four peptides from the top 17 candidates contain early glycative modifica-tions, which were not as frequently detected. Furthermore, eight peptides presented at least one stable glycoxidative or oxidative modification, often accompanied by oxidations on various amino acids. In contrast, eight potential AGEs were found on the candidate peptides, which is less than expected. These observations hint at a specific diversification of modifications compared to overall counts, indicating a potential for transient and spe-cific sugar- and oxidation-related changes.

For the correlation analysis (l. 353 to end of page):

Of the top 17 candidates, 114 and 151 very strongly (<0.8) correlate with t-tau and 56, 134, 2_166, and 31 with both t-tau and p-tau. Furthermore, eight additional peptides strongly (>0.6) correlate with t-tau and eight with p-tau.

In contrast, the median correlation with single amyloid-beta measures is more di-verse and very weak in Aβ 1-42 (-0.07, 0.06 for >=0.7), and no peptide from the top candidates shows a very strong correlation. In contrast, correlations to the Aβ ratio are primarily weak to moderate and negative (median -0.32, -0.45 for >=0.7) and more uniform. However, from the top candidates, peptides 125, 35, and 90 show a very strong negative correlation with the Aβ-ratio, and nine peptides are strongly correlated.

Of note, in our study, the age of the patients also correlated very strongly with t-tau (0.88) and strongly with the Aβ-ratio (-0.78) and p-tau (0.88) (see Suppl. Figure 1, Suppl. Data 3). Furthermore, peptides with AUCs >0.7 and only very weak to weak correlation with age correlate less strongly with the Aβ-ratio and tau markers but, on average, more strongly with Aβ 1-42. Two exemplary peptides for that finding are 150 and 131.

Peptide 203, a kyn-modified peptide of alpha-1-acid glycoprotein (ROC-AUC 0.8), showed correlations not resembling the high ROC-AUC candidates, e.g., negative correlations to Aβ 1-40 & 42 and p-tau.

  1. Ensure all figures and tables are clearly labeled and include comprehensive legends. For example, Figure 1 would benefit from a more detailed explanation of the types of modifications and their relevance. Consider adding a visual summary or flowchart of the study design to assist readers in understanding the workflow.

We added the following information in the description of Figure 1:

Glycative modifications comprise early modifications, where monosaccharides reacted with amino acids, forming Schiff bases of trioses, tetroses, pentoses and fructose (hexose) as well as the carbonyl glyoxal (glyoxal-imine). Intermediate modifications include carbonyl- and glu-cose-derived labile structures, which can react further to imidazolones and pyrraline (3-DG), CM-AGEs (Glarg/Glap), glucosepane (glucosone) or pentosidine (3-DG and pentosone), AGEs, which are glycative end products, include classical modifications, and the novel group of am-ide-AGEs which include oxidation-products of Amadori intermediates. Oxidative modifications are stable products of tryptophan residue oxidation.

We also added some information in the descriptions of Figs. 2 and 3.

Regarding the visual summary, we think that the graphical abstract (GA) prepared with the manuscript should assist readers sufficiently. Based on the text-revision, we also revised the GA.

  1. The discussion should more thoroughly explore the clinical implications of the findings. This inclu des how these peptide modifications could be used in early diagnosis or monitoring of AD and what further steps are needed to translate these findings into clinical practice.

We tried to include clinical relevance throughout the discussion, the main paragraph is:

From this point of view, modified peptide biomarkers, with glycative, glycoxidative or oxidative modifications, could be best suited to monitor the change or persistence of metabolic changes and oxidative conditions in the brain or at least CSF throughout the AD continuum similar to the clinical use of HBA1C in diabetes.

However, to enable clinical use of modified peptide markers, the findings would first need validation and further description of the occurrence of the peptides. Additionally, a precise, either MS or antibody-based detection method is needed to monitor specific modified peptides with a defined set of modification possibilities. This might be challenging with the current discovery-type workflow. As for favorable candidates, the identified peptide belonging to angiotensinogen would be of interest for this kind of validation because of its localization on the protein in an easily accessible unordered region and the knowledge generated about possible modifications from this study.

We also added a small paragraph to this end in the section about correlations:

Thus, our data supports the general notion of monitoring oxidative stress in patients with risk factors for neuro-degenerative diseases both by established methods like PET {Mota, 2022 #539}, and novel methods {Kehm, 2021 #540}.

And in the last section of the discussion:

From the current knowledge, it is more likely that the markers will be of use in identifying persons with a high-risk to develop neurodegenerative diseases in general.

  1. 13.   The conclusion should succinctly summarize the main findings and their significance, clearly stating whether the study supports the feasibility of using these peptide modifications as biomarkers. Highlight the need for larger, more comprehensive studies to validate these results.

We rephrased the conclusion to include the main findings and also statements regarding feasibility, limitation and further needed studies.

  1. 14.   Some references cited in the manuscript are older and might not represent the most recent advancements in the field. Updating the reference list to include more recent publications, especially those within the last five years, will enhance the manuscript’s relevance.

We thank the reviewer, for the careful review of the literature. We updated a large portion of the literature references to accommodate current data and knowledge during the manuscript revision. We hope, that the references now better reflect the relevance of the topic.

  1. Besides the limitations, the discussion should include suggestions for future research. This could involve validating findings in a larger cohort, exploring other potential biomarkers, or conducting longitudinal studies to assess changes over time.

We included a paragraph at the end of the discussion to include future research venues, as suggested:

In any case, future research is warranted to establish glycatively and oxidatively modified peptides as early biomarkers of AD. At first, the presented approach should be used in a larger cohort, including additional groups like patients at risk without cognitive impairment, MCI and different stages of AD to investigate the presence and magnitude of peptide-bound modifications along the AD continuum. To validate the candidates as di-agnostic markers, it would be necessary to determine the discrimination power against other types of dementia or mixed pathologies. From the current knowledge, it is more likely that the markers will help identify persons with a high risk of developing neuro-degenerative diseases in general. Especially for peptides detected in multiple modification states, it would be interesting to monitor those across the AD continuum and investigate a putative change evoked by the intensifying pathogenic mechanisms.

  1. The choice of nanoLC-LIT-Orbitrap-MS/MS for peptide analysis should be justified by discussing its advantages over other techniques in detecting modifications relevant to AD.

We absolutely agree with the reviewer – the instrumentation used here is definitely advantageous in analysis of protein Maillard reaction products, i.e. glycative and AGE-modifications. Low resolution instruments do not provide sufficient reliability of peptide identification and protein annotation. Identification of the modification type is also difficult to accomplish without a HR-MS instrument. Glycated protein adducts usually produce complex fragmentation patterns, which are better inter-pretable when softer fragmentation in a 3D quadrupole trap applied. The combination of these conside-rations led us to the decision to employ LIT-Orbitrap-MS.

We also added a corresponding paragraph of text in the discussion about the use of the instrumentation (l. 456 - 466):

For decades, high-resolution mass spectrometry with high-accuracy hybrid instruments like QqTOF-, Q-Orbitrap- and LIT-Orbitrap-MS has been recognized as the method of choice in nanoLC-based bottom-up-proteomics studies in general and Maillard-reaction product proteomics in particular {Frolov, 2010 #535;Cho, 2022 #536}. The quality of the resulting MS/MS spectra is critically important for identification, precise localization of the glycated site, and modification identity. Our previous studies indicated that softer collision conditions established in the HR mass analyzers are associated with higher reliability of peptide identification and yielded high identification rates for both early and ad-vanced glycated peptides and proteins {Frolov, 2014 #56;Greifenhagen, 2015 #72;Milkovska-Stamenova, 2019 #537}. Thus, LIT-Orbitrap-MS is preferable for the analysis of modified peptides.

  1. Provide a more detailed interpretation of the correlations between peptide modifications and established biochemical markers. Discuss what these correlations might imply about the underlying disease mechanisms.

As suggested, we provided a more detailed discussion on the correlation and the underlying mechanisms. For a more detailed description of results, please see point 10.

The discussion on correlation and the confounding factors now reads.

Nevertheless, the analysis of correlations between diagnostic biochemical core biomarkers and peptides hinted towards underlying mechanisms being reflected in the candidates. From the top candidates, it becomes clear that some peptides most strongly correlate with t-tau and p-tau, linking them to the tau-pathology, while others most strongly correlate with Aβ ratio. The second group of peptides might be interesting as putative biomarkers since the ratio was already shown to correlate negatively with cognitive decline and Aβ-PET scans independent of diagnosis [59]. It would be interesting to investigate whether specific types of modifications are associated with those different pathologies in a larger cohort.

One peptide from alpha-a1-glycoprotein modified with kynurenine (203 did not correlate strongly with age and showed distinctly different correlations from the other top-candidate peptides. This might be attributable to the proteins' role as acute-phase protein response to inflammation. At the same time, it is an example that oxidative stress can occur independently since kynurenine and other oxidative modifications were found on a range of peptides with different correlations, e.g., 151 with very high correlations to t-tau, 35 with strong correlations to Aβ-ratio and 203 mentioned before. Thus, our data supports the general notion of monitoring oxidative stress in patients with risk factors for neuro-degenerative diseases both by established methods like PET {Mota, 2022 #539}, and novel methods {Kehm, 2021 #540}.

Comments on the Quality of English Language

The manuscript is generally well-written, but some sentences could be rephrased for clarity and flow. A minor round of editing to correct grammatical errors and improve sentence structure would enhance readability.

Of course, we edited and corrected grammatical errors throughout the revision. Additionally, we used the software Grammarly to ensure proper use of language.

Submission Date

15 August 2024

Date of this review

24 Aug 2024 16:28:03

Round 2

Reviewer 2 Report

Comments and Suggestions for Authors

The authors responded adequately to my previous comments and substantially improved their manuscript so I can conclude for acceptance.